# Diguanylate Cyclase GdpX6 with c-di-GMP Binding Activity Involved in the Regulation of Virulence Expression in *Xanthomonas oryzae* pv. *oryzae*

**DOI:** 10.3390/microorganisms9030495

**Published:** 2021-02-26

**Authors:** Weiwei Yan, Yiming Wei, Susu Fan, Chao Yu, Fang Tian, Qi Wang, Fenghuan Yang, Huamin Chen

**Affiliations:** 1State Key Laboratory for Biology of Plant Diseases and Insect Pests, Institute of Plant Protection, Chinese Academy of Agricultural Sciences, Beijing 100193, China; yww_vickey@163.com (W.Y.); weiym132@163.com (Y.W.); yuchao@caas.cn (C.Y.); tianfang@caas.cn (F.T.); chenhuamin@caas.cn (H.C.); 2The MOA Key Laboratory of Plant Pathology, Department of Plant Pathology, College of Agronomy and Biotechnology, China Agricultural University, Beijing 100193, China; wangqi@cau.edu.cn; 3Shandong Provincial Key Laboratory of Applied Microbiology, Ecology Institute, Shandong Academy of Sciences, Jinan 250014, China; 1986fansusu@163.com

**Keywords:** *Xanthomonas oryzae* pv. *oryzae* 1, virulence 2, c-di-GMP 3, diguanylate cyclase 4, GGDEF domain 5

## Abstract

Cyclic diguanylate monophosphate (c-di-GMP) is a secondary messenger present in bacteria. The GGDEF-domain proteins can participate in the synthesis of c-di-GMP as diguanylate cyclase (DGC) or bind with c-di-GMP to function as a c-di-GMP receptor. In the genome of *Xanthomonas oryzae* pv. *oryzae* (*Xoo*), the causal agent of bacterial blight of rice, there are 11 genes that encode single GGDEF domain proteins. The GGDEF domain protein, PXO_02019 (here GdpX6 [GGDEF-domain protein of *Xoo*
6]) was characterized in the present study. Firstly, the DGC and c-di-GMP binding activity of GdpX6 was confirmed in vitro. Mutation of the crucial residues D^403^ residue of the I site in GGDEF motif and E^411^ residue of A site in GGDEF motif of GdpX6 abolished c-di-GMP binding activity and DGC activity of GdpX6, respectively. Additionally, deletion of *gdpX6* significantly increased the virulence, swimming motility, and decreased sliding motility and biofilm formation. In contrast, overexpression of GdpX6 in wild-type PXO99^A^ strain decreased the virulence and swimming motility, and increased sliding motility and biofilm formation. Mutation of the E^411^ residue but not D^403^ residue of the GGDEF domain in GdpX6 abolished its biological functions, indicating the DGC activity to be imperative for its biological functions. Furthermore, GdpX6 exhibited multiple subcellular localization in bacterial cells, and D^403^ or E^411^ did not contribute to the localization of GdpX6. Thus, we concluded that GdpX6 exhibits DGC activity to control the virulence, swimming and sliding motility, and biofilm formation in *Xoo***.**

## 1. Introduction

The phytopathogen *Xanthomonas oryzae* pv. *oryzae* (*Xoo*) causes the bacterial leaf blight in rice, one of the most devastating bacterial diseases of rice [1]. To infect a plant, *Xoo* mainly invades the rice through the wound or water pores, colonizes the xylem vessels of rice leaves [1]. The bacterial leaf blight disease causes different degrees of reduction rice yield with 70% yield reduction in the most severe cases [2,3]. The successful *Xoo* infection of a rice plant depends on several protein secretion systems, such as the type II secretion system that plays important roles in the degradation of plant cell walls and the type III secretion system which transfers the effectors into the plant cells [4,5,6]. In addition, multiple virulence factors like exopolysaccharide (EPS), biofilm, extracellular enzymes, and adhesins contribute to the virulence of *Xoo* [4]. These virulence factors have been found to be regulated by several signaling systems, including the two-component systems, diffusible signal factor signaling pathway, and cyclic diguanylate monophosphate (c-di-GMP) signaling pathway in *Xoo* [7,8,9,10,11].

c-di-GMP has been recognized as an essential secondary messenger that regulates biofilm formation, motility, cell differentiation, and virulence of pathogenic bacteria [12,13,14,15]. Diguanylate cyclase (DGC) with a GGDEF domain and phosphodiesterase (PDE) with an EAL or HD-GYP domain are responsible for the synthesis and degradation of c-di-GMP, respectively [16,17]. In general, high levels of c-di-GMP in bacteria can promote biofilm formation and switching of bacterial lifestyle from motile to sessile. However, low concentrations of c-di-GMP stimulate the motility and the expression of virulence factors [16,17]. Identification of multiple receptors can help to explain how c-di-GMP elicits specific responses. Several studies have revealed that c-di-GMP exerts its multiple regulatory roles by binding to RNA, various effectors, and regulatory proteins, including PilZ domain proteins, transcriptional regulators, degenerate GGDEF or EAL domain proteins, polynucleotide phosphorylase, riboswitches, and kinases, to name a few [18].

Many bacterial genomes encode dozens of GGDEF-domain proteins, and some of these are characterized by biochemical characteristics, molecular structure, and physiological functions. It has been reported that an active site (A-site) of GGDEF domain is crucial for the GGDEF-domain proteins to catalyze the condensation of two GTP molecules to form a molecule of c-di-GMP [16,17]. Additionally, the RXXD motif of the allosteric inhibition site (I site), which is located before the A site in GGDEF domain participates in c-di-GMP binding [19]. Several proteins contain a conserved A site in GGDEF domain function as DGCs, such as PleD of *Caulobacter crescentus*, WspR of *Pseudomonas aeruginosa* and the GGDEF-domain protein YeaP of *Escherichia coli* [20,21,22,23]. There is a general regulatory mechanism of some active DGCs including PleD and WspR, which allows them to bind c-di-GMP via the I site to feedback inhibit its DGC activity. Nonetheless, there are some exceptions, for instance, GGDEF-domain proteins with degenerated GGDEF motif including VCA0965 of *Vibrio cholerae* and ECA3270 of *Pectobacterium atrosepticum* catalyze the synthesis of c-di-GMP [24,25]. Besides the role of the I-site in DGC feedback inhibition, the non-catalytic GGDEF domain with degenerate A-site motifs including PopA in *C. crescentus* and SgmT in *Myxococcus xanthusmmi* can exert their regulatory roles by conserved c-di-GMP binding via the I site and represents an important class of c-di-GMP effector proteins [26,27].

In *Xoo*, c-di-GMP signaling plays important roles in regulating the expression of virulence factors [7,8,9,10,11]. The genome of *Xoo* PXO99^A^ codes for eleven proteins with single GGDEF domain [28]. So far, only two have been characterized: GdpX1, which is involved in the regulation of virulence, EPS production, and swimming motility, and DgcA, an active DGC which negatively regulates the pathogenicity of *Xoo* on rice, EPS production, motility, and auto-aggregation by production of c-di-GMP [10,29]. In the present study, PXO_02019 (here GdpX6 [GGDEF-domain protein of *Xoo*
6]), which contains an extracellular domain Cache_1, a transmembrane domain (TM domain) and a GGDEF domain was characterized. *In vitro* analysis revealed that GdpX6 is not only an active DGC, but also binds to c-di-GMP with low affinity. Furthermore, the present study provides evidence that GdpX6 controls the virulence, swimming and sliding motility, and biofilm formation in *Xoo*. Subcellular localization analysis identified that GdpX6 protein has multisite distribution in the cell. These findings demonstrate that GdpX6 functions as a DGC to negatively regulate the virulence in *Xoo*.

## 2. Materials and Methods

### 2.1. Bacterial Strains Plasmids and Growth Condition

The bacterial strains and plasmids used in this study are list in Table 1. *E. coli* strains were cultured in Luria–Bertani (LB) medium at 37 °C. *Xoo* wild-type strain PXO99^A^, derived mutants and complementary strains were cultured on peptone sucrose agar (PSA) medium or M210 liquid medium with appropriate antibiotics at 28 °C [30]. Relevant antibiotics were used at the following concentrations: gentamycin, 20 µg/mL; kanamycin (Km), 50 µg/mL; and ampicillin (Ap), 100 µg/mL [31].

### 2.2. Bioinformatics Analysis of GdpX6

Nucleotide and amino acid sequences were downloaded from National Center for Biotechnology Information (NCBI) (https://www.ncbi.nlm.nih.gov/pmc/) (accessed on 10 January 2021). The domain structures of GdpX6 were analyzed by the Simple Modular Architecture Research Tool (SMART) (http://smart.embl-heidelberg.de/) (accessed on 10 January 2021). Multiple-sequence alignments were performed using the DNAMAN software (version 5.2.2, Lynnon BioSoft, Sanramon, CA, USA).

### 2.3. Protein Expression and Purification

The truncated DNA fragment GdpX6^GGDEF^ encoding the intact GGDEF domain of GdpX6 was amplified by using the GdpX6PF1/GdpX6PR1 primer pairs. Point mutations of GdpX6 were generated by Bridge PCR, as described previously [8]. For point mutation of D^403^ in the GGDEF domain of GdpX6, the upstream and downstream fragments were amplified using primers GdpX6PF1/D^403^R, and D^403^F/GdpX6PR1. Subsequently, the fragments GdpX6^GGDEF-D403A^ were amplified using two fragments as the templates by the GdpX6PF1/GdpX6PR1 primers. Point mutation for E^411^ was constructed using the methods as that of GdpX6^GGDEF-D403A^. The primers used in the study are listed in Appendix A. The fragments GdpX6^GGDEF^, GdpX6^GGDEF-D403A^, and GdpX6^GGDEF-E411A^ were subjected to treatment with corresponding restriction endonuclease, followed by their insertion into the pColdSUMO plasmid, resulting in pCGdpX6^GGDEF^, pCGdpX6^GGDEF-D403A^, pCGdpX6^GGDEF-E411A^ constructs. The presence of the correct fragments GdpX6^GGDEF^, GdpX6^GGDEF-D403A^ and GdpX6^GGDEF-E411A^ was confirmed by sequencing. The plasmids were transferred into *E. coli* strains BL21(DE3) for protein expression. Expression of target proteins was induced by addition of isopropyl β- d-1-thiogalactopyranoside (IPTG) at the final concentration of 0.5 mM. Then, the bacterial cultures were incubated for 16 h at 16 °C. And the target proteins were purified as described previously [8,36]. At the same time, purified SUMO-His_6_ protein was purified as a negative control.

### 2.4. DGC Activity Assay

DGC activity of GdpX6 was firstly verified by using the riboswitch-based dual-fluorescence reporter system as described previously by Zhou et al. [34]. The constructs pCGdpX6^GGDEF^, pCGdpX6^GGDEF-E411A^ and the empty vector pColdSUMO were transformed to *E. coli* BL21(DE3) strain containing a triple-tandem riboswitch BC3-5 RNA. The *E. coli* strains containing pETPleD/BC3-5 RNA and pET28b/BC3-5 RNA were used as positive control and its corresponding negative control, respectively [34]. All bacterial strains were grown in LB medium at 28 °C until OD_600_ reached approximately 0.8. Thereafter 1 mM IPTG was added to the bacterial culture for 20 h for induction of the protein expression. The culture was then maintained at 4 °C for subsequent experiments. Samples were diluted to an OD_600_ of 0.1 with water, and the ratio of fluorescence intensity (RFI) was detected by Flex station 3 (Moleculer Devices, Sunnyvale, CA, USA).

The DGC activity of the proteins was further confirmed by using GTP as the substrate [37]. Around 100 µg target purified proteins were added into a reaction buffer (75 mM Tris-HCl (pH 7.8), 250 mM NaCl, 25 mM KCl, 10 mM MgCl_2_) containing 100 μM GTP. The mixture was incubated at 37 °C for 12 h. The reaction was terminated by heating the reaction mixture at 95 °C for 5 min, followed by centrifugation at 12000 rpm for 5 min. The c-di-GMP production in supernatant was analyzed by liquid chromatography-tandem mass spectrometry (LC-MS/MS).

### 2.5. ITC Isothermal Titration Calorimetry (ITC) Assay

The binding of the GdpX6 with c-di-GMP was performed on an ITC200 calorimeter (MicroCal, Northampton, MA). Briefly, the proteins (30 μM SUMOHis_6_, 10 μM SUMOHis_6_-GdpX6^GGDEF^, 10 μM SUMOHis_6_-GdpX6^GGDEF-D403A^) were syringed into the cell pool and 2 μL c-di-GMP solution (300 μM, 1 mM, 1 mM) as ligand was titrated with the sample at 120 sec intervals with stirring speed of 1000 rpm at 25 °C. The heat changes accompanying these additions were recorded. The data of the titration experiment was calibrated with a buffer control. The dissociation constant (*Kd*) was analyzed through the single-site model using the MicroCal ORIGIN version 7.0 software [35].

### 2.6. Construction of Gene Deletion Mutant, Complementation and Overexpression Strains

The Δ*gdpX6* mutant was generated using the suicide vector pKMS1 by homologous recombination, as described previously [38]. The left and right arms of *gdpX6* were amplified by PCR using PXO99^A^ genomic DNA as template with specific primers (Appendix A). The fragments were digested and subsequently ligated into the pKMS1vector to generate the pK*gdpX6* construct. The plasmid was transformed into PXO99^A^ by electroporation and subsequently screened on NAS plate with 10% sucrose. The single colonies that were resistant to 10% sucrose but sensitive to Km were further confirmed as mutant by PCR analysis.

The coding region of *gdpX6* with ribosome-binding site but without the termination codon was amplified with gdpX6CF/gdpX6CR primers and inserted into broad-host-range pBBRMCS1 plasmid containing a *gfp* gene which encoded a green fluorescent protein (GFP), resulting in pB*gdpX6gfp* plasmid. Point mutation of D^403^ and E^411^ of GdpX6 were constructed using the method described above, resulting in the plasmids pB*gdpX6^D403A^gfp* and pB*gdpX6^E411A^gfp*. pB*gdpX6gfp*, pB*gdpX6^D403A^gfp* and pB*gdpX6^E411A^gfp.* These plasmids were then transformed into ∆*gdpX6* as well as PXO99^A^ by electroporation and screened on PSA plate containing 100 μg/mL ampicillin antibiotic to acquire complementation and *gdpX6*-overexpressing strain, respectively. The expression of GFP-fusion proteins was confirmed by western blotting. The primers used in the study are listed in Appendix A.

### 2.7. Western Blotting Analysis

For analysis of the expression of GFP-fusion proteins in PXO99^A^, *Xoo* strain was cultured in M210 medium at 28 °C to OD_600_ of 1.0. Bacteria cells were collected by centrifugation at 12,000 rpm for 5 min. The cells were resuspended in phosphate-buffered saline (PBS) and ultrasonicated for 2 min. The samples were boiled, separated on 12% SDS-PAGE and subsequently transformed onto polyvinylidene fluoride membranes (Merck Millipore, Darmstadt, Germany) for immunoblotting using anti-GFP primary antibodies (Huaxingbio, Beijing, China). Goat anti-mouse secondary antibody conjugated with HRP (horseradish peroxidase) were used to recognize the primary antibodies (TransGen Biotech, Beijing, China). Enhanced HRP-DAB Chromogenic Kit (TransGen Biotech, Beijing, China) was used to check the target proteins on the membrane according to the manufacturer’s guidelines.

### 2.8. Virulence Assay

The pathogenicity assay on susceptible rice cultivar (*Oryza sativa* L. cv. Nipponbare) was used to assess the virulence of wild-type PXO99^A^ and its derived strains [8]. Firstly, all strains were cultured in M210 medium at 28 °C to OD_600_ of 0.8–1.0. Bacterial cells were harvested by centrifugation and subsequently resuspended in the same volume of sterilized ddH_2_O. The bacterial cells were inoculated onto the rice leaves by the leaf clipping method. At least ten leaves were inoculated for each strain in each experiment. After 14 days, the lesion length was measured, and the leaves were photographed. The experiment was repeated three times.

### 2.9. Motility Assay

*Xoo* strains were grown in M210 at 28 °C until OD_600_ reached 0.8. Thereafter, bacteria cells were harvested and resuspended in ddH_2_O. A total of 2 μL of bacterial suspension was stabled into semi-solid plates containing 0.25% agar for testing swimming assay or SB medium plates containing 0.6% agar for analyzing sliding motility, respectively [39,40]. Pictures were taken after bacterial growth at 28 °C for 4 days. The diameters of the swimming or sliding zones were measured. The experiment was repeated thrice in triplicate.

### 2.10. Biofilm Formation Assay

*Xoo* cells were grown in M210 and diluted in M210 to an OD_600_ of 0.5. 200 μL of the cultures were transferred to a 96-well polystyrene microplate and incubated at 28 °C for 4 days. The medium was discarded, and biofilm was washed with distilled water. The biofilm was subjected to staining with 0.1% crystal violet for 15 min [41]. The biofilm was washed twice with distilled water and photographed. For quantification, the biofilm was dissolved in ethanol, and the absorbance was detected at 490 nm with Flex station 3 (Molecule Devices, Sunnyvale, CA, USA). All experiments were performed thrice in triplicate.

### 2.11. EPS Production Assay

The ethanol precipitation method was used to quantify the EPS production of *Xoo* strains [42]. The *Xoo* strain was cultured in M210 medium until the OD_600_ reached 2.5. The supernatants were collected by centrifugation at 6,000 rpm for 10 min. Supernatants were subsequently incubated at −20 °C overnight following the addition of two volumes of absolute ethanol to the supernatants. The EPS molecules were collected by centrifugation at 10,000 rpm for 20 min and then dried overnight at 55 °C. The weights of EPS molecules were determined. At the same time, 2 μL of bacterial supernatant was stabbed onto a PSA plate and incubated at 28 °C for 3–4 days and then photographed. All experiments were performed thrice in triplicate.

### 2.12. Extracellular Enzymatic Activities Assay

Extracellular enzymatic activities of *Xoo* strains were tested as described previously [43]. Bacterial cells were cultured in M210 at 28 °C until an OD_600_ of 0.8 was reached. Thereafter, 2 μL cultures were stabbed onto PSA plate with 0.2% RBB-xylan. Xylanase activity was detected by the appearance of white clear zones against a blue background on the plate after incubation at 28 °C for 2 days. On the other hand, 2 μL cultures were stabbed onto PSA plate with 0.5% carboxymethyl cellulose. Plates were cultured at 28 °C for 2 days. The plates were subsequently stained with 1% Congo red for 30 min. Thereafter, the plates were washed twice with 1 M NaCl solution for 20 min. Cellulase activity was observed by the appearance of the transparent circle under the red background. All experiments were performed thrice in triplicate.

### 2.13. Fluorescence Microscopy

The PXO99^A^(pB*gdpX6gfp*) and PXO99^A^(pB*gdpX6^D403A^gfp*), PXO99^A^(pB*gdpX6^E411A^gfp*) strains were grown in M210 medium to an OD_600_ of 1.0. The cell suspension was dripped on a glass slide and covered with a coverslip. The subcellular location signals of samples were observed and recorded using an Olympus BX61 microscope (Olympus, Tokyo, Japan).

### 2.14. Statistical Analysis

All analysis was performed using Microsoft Excel 2010. The means and standard deviations of experimental results were calculated using average function and STDEV function, respectively. And *t* test was used to determine significant differences between samples.

## 3. Results

### 3.1. GdpX6 Contains a Conserved GGDEF Domain

GdpX6, one of eleven genes encoding single GGDEF domain proteins in the genome of *Xoo* PXO99^A^, contains an extracellular Cache_1 protein domain (41 to 275 amino acids [aa]) that is predicted to have a role in small-molecule recognition [44], a TM domain (294 to 311 aa), and a GGDEF domain (318 to 495 aa) (Figure 1a). Blast searches revealed that GdpX6 has homologous proteins in several sequenced *Xanthomonas* species genomes (Appendix A). GdpX6 was found to be homolog of XCC2731 in *X. campestris* pv. *campestris* (*Xcc*), XOC1515 in *X. oryzae* pv. *oryzicola* (*Xoc*) and XAC2897 in *X. citri pv. citri* (*Xac*), with the amino acid sequence similarities of 78.93%, 92.47%, and 65.01%, respectively. (Appendix A). Previous studies have revealed the capacity of GGDEF domain to synthesize c-di-GMP or bind to c-di-GMP depends on the presence of A-site and I-site in GGDEF domain, respectively [16,17]. The amino acid sequence alignment with the reported active DGCs indicated that GGE^411^EF residues of A-site, RXXD^403^ residues of I-site, Mg^2+^, and GTP binding sites were conserved in GdpX6 (Figure 1b). Thus, GdpX6 might function as an active DGC and bind with c-di-GMP.

### 3.2. GdpX6 Demonstrates DGC Activity In Vitro

To evaluate whether GdpX6 possesses the DGC activity, the protein GdpX6^GGDEF^ containing the GGDEF motif of GdpX6 and the mutagenized protein GdpX6^GGDEF-E411A^ having mutation in important residue E^411^ of the GGDE^411^F motif for the catalysis of diguanylate cyclase, were expressed, respectively. Firstly, the DGC activity of GdpX6 was measured using the dual-fluorescence reporter system by visible fluorescence color changes and the RFI values in *E. coli* BL21(DE3) [34]. The results showed that the color of bacterial cells containing pCGdpX6^GGDEF^ expression vector changed from green to red and the relative RFI values significantly increased about 13 folds as compared to the bacterial cells containing the negative vector pColdSUMO after 20 h IPTG induction (*p* < 0.05) (Figure 2a). These results were similar to those of the bacterial cells containing PleD from *C. crescentus,* which exhibited strong DGC activity compared to its negative control pET28b [34] (Figure 2a). The bacterial cells containing point mutant pCGdpX6^GGDEF-E411A^ expression vectors showed the same color as that of the negative pColdSUMO control. However, the value of RFI decreased about 81% as compared to GdpX6^GGDEF^ (Figure 2a), suggestive of the lower DGC activity of GdpX6^GGDEF-E411A^ as compared to GdpX6^GGDEF^. Subsequently, the enzyme activity of purified SUMOHis_6_-GdpX6^GGDEF^, SUMOHis_6_-GdpX6^GGDEF-E411A^ and SUMOHis_6_ proteins was tested using GTP as substrate and the concentration of c-di-GMP in the reaction was detected by LC-MS/MS. As shown in Figure 2b, approximately 600 ng/mL c-di-GMP was synthesized by the protein SUMOHis_6_-GdpX6^GGDEF^, while no c-di-GMP synthesis was detected in the negative SUMOHis_6_ control. SUMOHis_6_-GdpX6^GGDEF-E411A^ did not exhibit the wild-type DGC activity (Figure 2b). These results suggest that GdpX6 is an active DGC.

### 3.3. GdpX6 Binds to c-di-GMP via the I Site of GGDEF Motif

The bioinformatic analysis demonstrated that GdpX6 contains a conserved I-site that can bind to c-di-GMP [45]. To identify whether GdpX6 binds to c-di-GMP, the c-di-GMP binding affinity of the proteins GdpX6, GdpX6^GGDEF-D403A^ and SUMOHis_6_ was determined by ITC. The results showed that GdpX6^GGDEF^ bound c-di-GMP with the dissociation constants (*Kd*) of 9 ± 2.98 µM (Figure 3b), while no interaction was detected between the control protein SUMOHis_6_ and c-di-GMP (Figure 3a). To further characterize the importance of I site of GdpX6, the D^403^ residue of I site within the protein was mutated, and the resultant mutant protein was evaluated for its c-di-GMP binding ability. It was found that the mutated protein GdpX6^GGDEF-D403A^ failed to bind to c-di-GMP (Figure 3c). These results indicate that GdpX6 binds c-di-GMP through I-site.

### 3.4. GdpX6 Contributes to the Virulence of Xoo on Rice

Several GGDEF-domain proteins have been shown to be involved in the regulation of bacterial virulence in *Xanthomonas* species [10,29,46]. To investigate whether GdpX6 affects the virulence of *Xoo*, a *gdpX6*-deleted strain, complement strains of ∆*gdpX6* and *gdpX6*-overexpressing strains were constructed. The expressions of GdpX6-GFP, GdpX6^D403A^-GFP and GdpX6^E411A^-GFP in wild-type PXO99^A^ and ∆*gdpX6* were confirmed at expected sizes by western blotting analysis (see Appendix A). The bacterial cells were inoculated onto the leaves of susceptible rice plants by leaf-clipping method. The disease symptoms of rice were measured and recorded after 14 days of inoculation. As shown in Figure 4a,b, the mutant ∆*gdpX6* strain caused more severe disease symptoms and the lesion lengths were increased by about 20% as compared to wildtype strain PXO99^A^. *In trans* expression of the full length *gdpX6*-*gfp* in ∆*gdpX6* restored the phenotype to near-wild-type levels. The results showed that expression of point mutation protein GdpX6^D403A^-GFP in ∆*gdpX6* restored the virulence to near-wild-type levels, while expression of GdpX6^E411A^-GFP in ∆*gdpX6* showed similar virulence to that of the mutant ∆*gdpX6* strain (Figure 4a,b). These results demonstrate that the GFP fusion proteins of GdpX6 retained their function. Moreover, overexpression of GdpX6-GFP or GdpX6^D403A^-GFP in PXO99^A^ significantly decreased the virulence (by about 25%) in comparison to PXO99^A^ (*p* < 0.05), while as overexpression of GdpX6^E411A^-GFP in PXO99^A^ failed to inhibit the virulence of *Xoo*. These results suggest that GdpX6 negatively regulates the bacterial virulence in *Xoo* and the GGDEF domain of GdpX6 plays a primary role in regulation of virulence in rice.

### 3.5. GdpX6 Is Involved in the Regulation of Swimming and Sliding Motility of Xoo

To investigate whether *gdpX6* influences the motility of *Xoo*, the wild-type strain PXO99^A^, the mutant ∆*gdpX6* strain and the complement strains of ∆*gdpX6* and *gdpX6*-overexpressing strains were cultured on semi-solid plates containing 0.25% agar for flagellum-dependent swimming motility or SB medium containing 0.6% agar for type IV pilis-dependent sliding motility as described in previous studies [47]. As shown in Figure 5a, ∆*gdpX6* displayed larger swimming zones, and the swimming diameters increased to about 3 mm than PXO99^A^. The complement strains Δ*gdpX6* (pB*gdpX6gfp*) and Δ*gdpX6*(pB*gdpX6^D403A^gfp*) recovered the swimming motility near to that of wild-type, while Δ*gdpX6*(pB*gdpX6^E411A^gfp*) showed similar swimming motility to the Δ*gdpX6* strain (Figure 5a). Overexpression of *gdpX6* in PXO99^A^ led to a significant decrease in the swimming motility of *Xoo* (*p* < 0.05) (Figure 5a). When the E^411^ or D^403^ were mutated in GdpX6, the influence of overexpression of *gdpX6* in PXO99^A^ on the swimming motility disappeared (Figure 5a). Moreover, results from the sliding motility assays showed that the sliding zones of Δ*gdpX6* mutant were smaller than that of PXO99^A^, while complement strains Δ*gdpX6*(pB*gdpX6gfp*) restored it to that of the wild-type (Figure 5b). The complement Δ*gdpX6*(pBg*dpX6^E411A^gfp*) strain showed a similar sliding zone as that of the Δ*gdpX6* mutant, while the Δ*gdpX6*(pB*gdpX6^D403A^gfp*) strain displayed similar sliding motility as that of the Δ*gdpX6*(pB*gdpX6gfp*) strain (Figure 5b). Overexpression of GdpX6 or GdpX6^D403A^ in PXO99^A^ resulted in the enlarged sliding motility as compared to the wild type, while expression *in trans* of GdpX6^E411A^ in the PXO99^A^ failed to enhance its sliding motility (Figure 5b). These results indicate that GdpX6 regulates swimming motility negatively but sliding motility positively in *Xoo*.

### 3.6. GdpX6 Promotes Biofilm Formation of Xoo

It has been reported that c-di-GMP affects the adhesion of bacteria to host cells via regulation of the biofilm formation in bacteria [48]. Therefore, we examined whether deletion and overexpression of *gdpX6* influence biofilm formation of *Xoo*. Results from biofilm formation assays showed that biofilm formation of Δ*gdpX6* mutant decreased by approximately 21 % relative to PXO99^A^, while complement strains Δ*gdpX6* (pB*gdpX6gfp*) restored the phenotype near to PXO99^A^ (Figure 6). The complement strain Δ*gdpX6* (pB*gdpX6^D403A^gfp*) restored biofilm formation similar to that of the Δ*gdpX6* (pB*gdpX6gfp*) strain, while Δ*gdpX6* (pB*gdpX6^E411A^gfp*) strain displayed a biofilm formation level similar to that of the Δ*gdpX6* mutant (Figure 6). In contrast, biofilm formation increased approximately 33% and 32% GdpX6 or GdpX6^D403A^ under overexpression in PXO99^A^ in comparison to PXO99^A^, while as there were no significant differences in terms of biofilm formation under GdpX6^E411A^ overexpression in PXO99^A^ (Figure 6). These results suggest that *gdpX6* promotes biofilm formation in *Xoo*, where the residues E^411^ are indispensable for the regulation of biofilm formation by GdpX6.

### 3.7. GdpX6 Does Not Control EPS Production and Extracellular Enzymatic Activities of Xoo

The regulation of GGDEF-domain proteins on EPS production has been reported in several bacteria [29,49,50]. Deletion of PDEs or DGCs encoding genes in *Xoo* has a significant impact on EPS production [8,11,29], so we compared the EPS production levels between wild-type strain, mutant ∆*gdpX6* strain, and *gdpX6*-overexpressing strain. Results from colony examining and EPS quantification showed that there were no differences between PXO99^A^, the mutant ∆*gdpX6* strain and *gdpX6*-overexpressing strain in terms of EPS production (see Appendix A). These results indicate that GdpX6 is not involved in the EPS production of *Xoo*.

The cellulase and xylanase activities contribute to the virulence of *Xoo* [43]. We analyzed whether GdpX6 affected the cellulase and xylanase activities of *Xoo*. No significant differences in cellulase and xylanase activities of PXO99^A^, the mutant ∆*gdpX6* strain and *gdpX6*-overexpressing strain were found (see Appendix A). These results indicate that GdpX6 does not regulate the extracellular enzymatic activities of *Xoo*.

### 3.8. Subcellular Localization of GdpX6 in Xoo

The subcellular localization of a protein is considered essential for the execution of its biological function in bacteria [51,52]. Therefore, we analyzed the subcellular localization of GdpX6-GFP, GdpX6^D403A^-GFP and GdpX6^E411A^-GFP proteins in PXO99^A^. In all bacterial cells tested, GdpX6-GFP, GdpX6^D403A^-GFP and GdpX6^E411A^-GFP proteins displayed multisite distributions of 96%, 96.0% and 93.4%, respectively, while less than 7% of the cells showed other subcellular location types including bipolar and unipolar (Figure 7). These results demonstrate that GdpX6 mainly exhibits multisite localization in the *Xoo* cells, and point mutants in the A or I site residues of the GGDEF domain do not influence the subcellular localization of GdpX6.

## 4. Discussion

GGDEF-domain proteins are widely distributed in bacterial genomes, many of these function as DGCs or receptors to regulate a diversity of biological phenotypes [53,54]. Previous studies have shown that GGDEF-domain proteins, including GdpX1 and DgcA, play crucial roles and participate in regulation of biological functions via c-di-GMP signaling in *Xoo* [10,29]. In this study, we functionally characterized the GGDEF-domain protein GdpX6, which was demonstrated to be an active DGC that negatively regulates the virulence and swimming motility, and positively regulates sliding motility and biofilm formation of *Xoo*. Therefore, GdpX6 is a novel DGC in *Xoo* and modulates multiple virulence-related phenotypes.

The homologs of GdpX6 exist in *Xanthomonas* species including *Xcc*, *Xoc,* and *Xac* (Appendix A). GdpX6 shows protein sequence similarity of 78.93% to XCC2731 from *Xcc*. XCC2731 regulates the aggregation of cells, motility, extracellular enzymes, and EPS production in *Xcc*, while the transcript level of *XCC2731* is regulated by Clp [46]. The present study did not find any role of *gdpX6* in EPS production and extracellular enzymes as reported for XCC2731. However, it was found that GdpX6 not only regulates motility and virulence but also controls biofilm formation and sliding motility in *Xoo*. It indicates that the homologs of GdpX6 from different species with high sequence similarity exhibit differences in the regulation of biological functions. Moreover, we confirmed that GdpX6 functions as novel DGC in *Xoo*. Based on the studies carried on the homologs of GdpX6 [46], our findings provide further information about special regulatory roles of GdpX6 in *Xoo* and new evidence on the mechanism of GdpX6 in the regulation of expression of virulence factors might be an DGC to affect the concentration of c-di-GMP in bacterial cells.

Structural analyses of GGDEF domains have revealed diverse mechanism for c-di-GMP synthesis and c-di-GMP binding [55]. DGCs have a conservative GG(D/E)EF motif in their A-site, which is crucial for the synthesis of c-di-GMP [16,17]. The RXXD^403^ residues of I-site in GGDEF motif are necessary for the c-di-GMP binding activity [19]. It is considered that some DGCs including PleD and WspR with conserved I-site are mainly involved in c-di-GMP synthesis. Moreover, the binding of DGC to c-di-GMP through I-site is to inhibit its DGC activity to maintain the balance of c-di-GMP level in vivo [20,21,22]. Bioinformatic analysis revealed that GdpX6 possesses both conserved A and I sites in GGDEF motif. We firstly confirmed that GdpX6 not only can synthetize the c-di-GMP but also binds to c-di-GMP in vitro. Point mutation in A or I site results in the loss of the activity of GdpX6. It suggests that GdpX6 might function as the reported DGCs with conserved I site like PleD or WspR [20,21,22]. Moreover, in vivo functional analysis demonstrated that E^411^ residue in A site but not D^403^ residue in I site of GdpX6 is essential for biological functions of GdpX6. These results of the present study suggest that GdpX6 mainly exerts its biological functions as a DGC. It is possible that the binding affinity of GdpX6 with c-di-GMP is not enough to affect its function in bacterial cells. However, the DGC activity of GdpX6 might be affected by binding to c-di-GMP in certain conditions as in the case of the PleD or WspR.

It is well established that DGCs play distinct roles in processes associated with motility, attachment, and biofilm formation [12,13]. Opposite regulation of swimming motility and biofilm formation by DGCs have been demonstrated. For example, deletion of DGCs EdcC and EdcE in *Erwinia. amylovora* increased swimming motility but decreased biofilm formation [56]. Moreover, flagellum-dependent biofilm regulatory response can be induced through the elimination of flagellum, which can improve the level of c-di-GMP and enhances the biofilm formation, and this response requires at least three specific DGCs in the *V*. *cholerae* [57]. In this study, we showed that deletion or overexpression of *gdpX6* in *Xoo* resulted in the opposite effects on biofilm formation and swimming motility that is consistent with the above active DGCs [56,57]. These findings revealed the role of DGCs in regulation of motile and static lifestyle in bacteria. Although EPS as a main component of biofilm is not regulated by GdpX6, it is possible that GdpX6 affects biofilm formation mainly by affecting flagellum-dependent motility of *Xoo*. The regulatory network that connects c-di-GMP signaling, motility, and biofilm formation in *Xoo* needs further investigation.

Type IV pili (T4P) are surface filamentous bacterial organelle involved in many phenotypes including adhesion, twitching or gliding motility, biofilm formation, and virulence in Gram-negative bacteria [58]. The c-di-GMP signaling and specific diverse c-di-GMP receptors have been demonstrated to be involved in the regulation of T4P. In *V. cholerae*, c-di-GMP could bind to ATPase MshE associated with mannose-sensitive haemagglutinin T4P formation, thus regulates pilus extension and retraction dynamics [59]. The c-di-GMP receptor FimXxcc and its interactor PilZxcc from *Xcc*, as well as its homologue Filp and PilZX3 from *Xoo*, regulate the bacterial T4P-depending sliding motility via direct interaction with or by affecting the expression of the pilus-related proteins, respectively [36,60,61,62]. On the other hand, T4P reversely regulates c-di-GMP signaling. PilR, a regulatory protein of pilus synthesis, controls the intracellular levels of c-di-GMP to inhibit production of the antifungal antibiotic HSAF in the soil bacterium *Lysobacter enzymogenes* [63]. In the present study, GdpX6 was found to act as a positive regulator of sliding motility of *Xoo*. This indicates that GdpX6 might be involved in the regulation of TP4. Previous studies have shown that the concentration of c-di-GMP plays an essential role in assembly of TP4. For example, the assembly of TP4 requires FimX with both c-di-GMP binding and PDE activity at low c-di-GMP concentrations, but this dependence disappeared at high c-di-GMP concentrations in *P. aeruginosa* [64,65]. Moreover, high intracellular c-di-GMP concentration increased the transcript levels of T4P genes as a result of its binding to an upstream transcriptionally active riboswitch in *Clostridium difficile* [66]. Therefore, we propose that GdpX6 might participate in regulation of sliding motility by affecting TP4 biogenesis or assembly as a DGC.

In the present study we observed that GdpX6-GFP displayed a multisite subcellular localization in *Xoo*, and both GdpX6^D403A^-GFP and GdpX6^E411A^-GFP exhibit similar multisite subcellular localization as that of GdpX6-GFP. The D^403^ is important for the c-di-GMP binding activity of GdpX6, and the E^411^ of GdpX6 is crucial for the DGC activity and the biological functions of GdpX6. It suggests that the DGC activity or c-di-GMP binding activity is not important for subcellular localization of GdpX6. These phenotypes are in agreement with other observations on subcellular localization of DGC or PDEs. For example, mutation of the residue G^368^ in the GGDEF motif of PleD, E^153^, and E^176^ in the EAL domain of EdpX1 abolish the DGC or PDE activity but do not affect the localization in the bacterial cell [11,52]. Moreover, the signal transduction domain of the protein can sense the specific signals and regulate the activity of DGC or PDE, thus affecting the subcellular localization of the protein. In *P. aeruginosa*, mutation of the phosphor-accepting residue D^70^ in the REC domain of WspR, which is essential for its DGC activity, alters the subcellular location of WspR from clustering to dispersion [67,68]. However, the TM domain which does not influence the function of the proteins has been shown to pay important roles in the subcellular localization of the proteins [11,69]. Besides the GGDEF domain, GdpX6 contains an N-terminal extracellular protein domain Cache_1 domain and a TM domain. Further identifying whether these domains are involved in the regulation of functions of GdpX6, and influence the subcellular location of GdpX6 in the cell will provide more information about the functional characteristics and localization patterns of GdpX6.

## 5. Conclusions

This study showed that the GGDEF-domain protein GdpX6 is an active DGC with c-di-GMP binding activity. Moreover, GdpX6 negatively regulated virulence and swimming motility, and positively regulated sliding motility and biofilm formation of *Xoo*. The A site of GGDEF domain is necessary for its regulatory functions in *Xoo*. Thus, we concluded that GdpX6 exists as a DGC to control the virulence expression in *Xoo*.

## Figures and Tables

**Figure 1 microorganisms-09-00495-f001:**
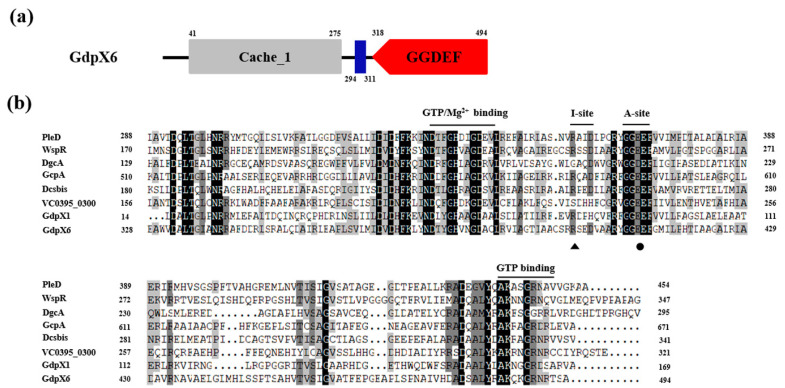
Domain organization and sequence alignment of GdpX6. (**a**) Schematic representation of the domain structures of GdpX6 (GenBank accession no. WP_012445692.1). The numbers represent the start and end amino acid of the predicted domains based on National Center for Biotechnology Information’s (NCBI’s) conserved domain database and SMART database. The Cache_1 domain (gray), transmembrane domain (blue) and GGDEF domain (red) are shown. (**b**) The amino sequence alignment of the GGDEF domain of GdpX6 with active DGCs, including PleD (GenBank accession no. AAA87378.1) from *C**. crescentus* CB15, WspR (GenBank accession no. NP_252391.1) from *P. aeruginosa* PAO1, GcpA (GenBank accession no. NP_460938.1a) from *Salmonella enterica* subsp. enterica serovar Typhimurium str. LT2, Dcsbis (GenBank accession no. NP_251461.1) from *P. aeruginosa* PAO1, Vc0395_0300 (GenBank accession no. ABQ19213.1) from *V. cholerae O395*, DgcA (GenBank accession no. AAW77242.1) from *Xoo* KACC10331, GdpX1 (GenBank accession no. WP_011259000.1) from *Xoo* PXO99^A^, by using the software DNAMAN. represents the putative residue confirmed by isothermal titration calorimetry assay involved in c-di-GMP binding and represents the putative residue confirmed by DGC activity assays involved in the synthesis of c-di-GMP. The amino acids highlighted in black and gray represent a homology level of 100% and ≥ 75%, respectively.

**Figure 2 microorganisms-09-00495-f002:**
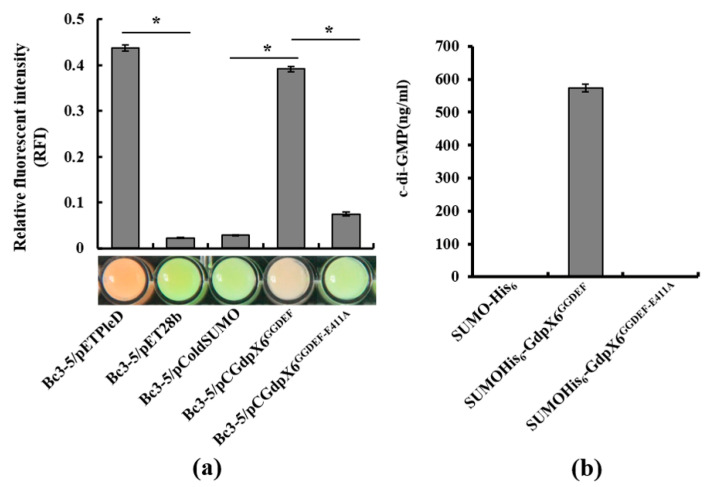
Analysis of diguanylate cyclase (DGC) activity of GdpX6. (**a**) Verification of the DGC activity of GdpX6 in *E. coli* BL21(DE3) using riboswitch (Bc3-5 RNA) based dual-fluorescence system. All bacteria strains were induced with 1 mM IPTG at 28 °C for 20 h. Visible fluorescence of bacterial suspensions was photographed. The relative fluorescence intensity (RFI) is calculated as the ratio of fluorescence intensity at 489 nm to fluorescence intensity at 547 nm. (**b**) The DGC activity of GdpX6 was detected by LC-MS-MS. Purified proteins were incubated with 100 μM of GTP overnight. HPLC was carried out to analyze the products. Three independent experiments were carried out with similar results. * indicates *p* < 0.05 as determined by *t*-test.

**Figure 3 microorganisms-09-00495-f003:**
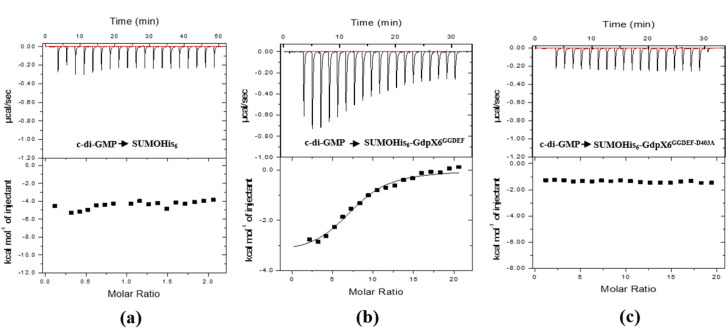
The binding activity of the GdpX6 with c-di-GMP were analyzed using isothermal titration calorimetry assays. The titration calorimetry of 30 μM SUMOHis_6_, 10 μM SUMOHis_6_-GdpX6^GGDEF^, and 10 μM SUMOHis_6_-GdpX6^GGDEF-D403A^ were syringed into the cell pool and 2 μL aliquots of 300 μM, 1 mM, or 1 mM c-di-GMP at 25 °C. The c-di-GMP binding to the control protein SUMOHis_6_ (**a**), the recombinant proteins SUMOHis_6_-GdpX6^GGDEF^ (**b**) and SUMOHis_6_-GdpX6^GGDEF-D403A^ (**c**).

**Figure 4 microorganisms-09-00495-f004:**
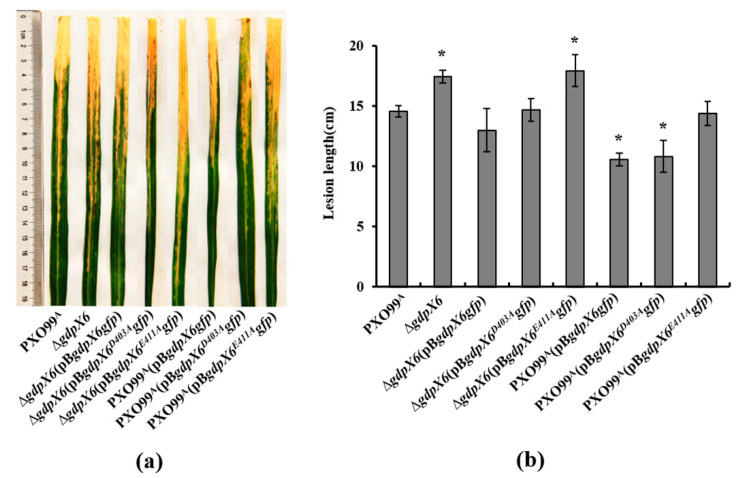
GdpX6 inhibited the virulence of *Xoo* on rice. (**a**) The bacterial cells were tested on rice by the leaf-clipping method. The disease symptoms were observed at 14 days post-inoculation. (**b**) The lesion lengths were recorded. The error bars represented standard deviations of the lesion lengths from ten leaves. Three independent experiments were performed with similar results. * indicates *p* < 0.05 as determined by *t*-test.

**Figure 5 microorganisms-09-00495-f005:**
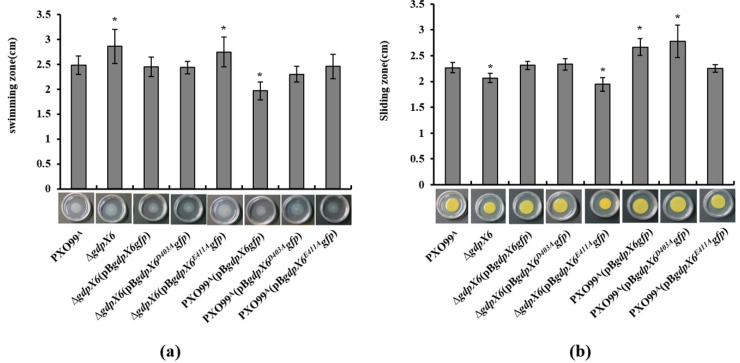
GdpX6 regulated swimming motility and sliding motility of *Xoo*. *Xoo* strains were inoculated on (**a**) semi-solid plates containing 0.25% agar for testing the swimming motility and (**b**) Super Broth (SB) plates containing 0.6% agar for testing the sliding motility at 28 °C for 4 days. The values are the means ± standard deviations from three replicates of three independent experiments. * indicates *p* < 0.05 as determined by *t*-test.

**Figure 6 microorganisms-09-00495-f006:**
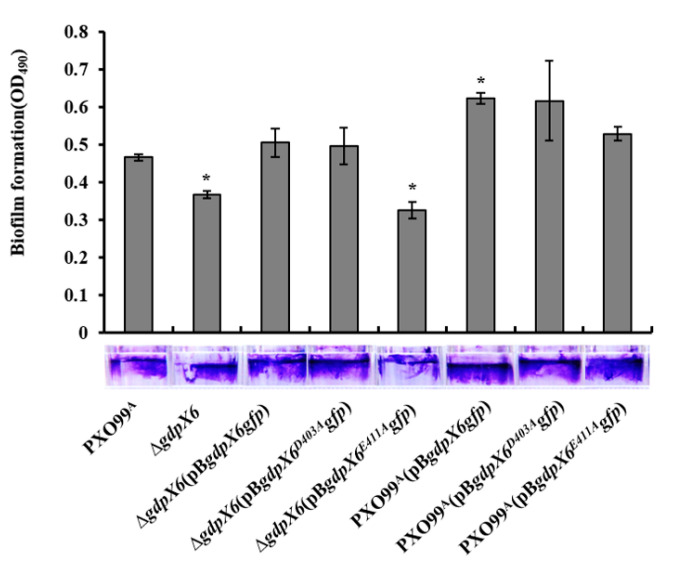
GdpX6 promoted biofilm formation of *Xoo*. Biofilm formation of all *Xoo* strains were tested by crystal violet staining method in polystyrene 96-well microplates. After suspension in ethanol, biofilm was quantified by measuring the optical density at 490 nm. Error bars represent standard deviations from three biological repeats. * indicates *p* < 0.05 as determined by *t*-test.

**Figure 7 microorganisms-09-00495-f007:**
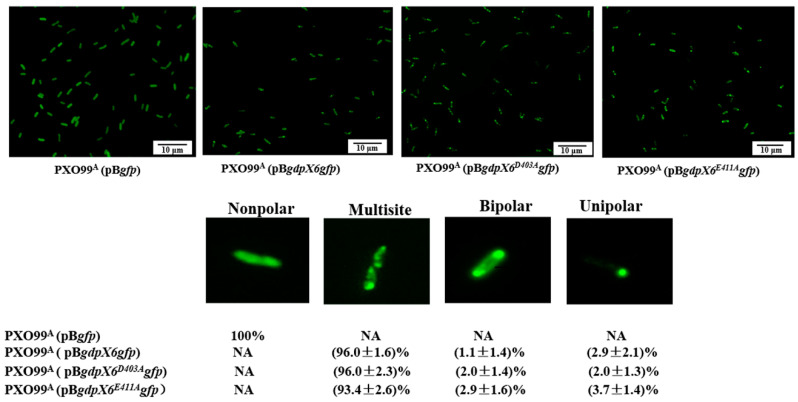
Subcellular localization of GdpX6 in *Xoo* PXO99^A^. PXO99^A^ (pB*gfp*), PXO99^A^ (pB*gdpX6**gfp*), PXO99^A^ (pB*gdpX6^D403A^gfp*) and PXO99^A^ (pB*gdpX6^E411A^gfp*) were grown in M210 to an OD_600_ of 1.0. Photograph of the GFP fusion protein in PXO99^A^ detected using a fluorescence microscope (Olympus BX61). Different subcellular locations of proteins GdpX6-GFP, GdpX6^D403A^-GFP, and GdpX6^E411A^-GFP including nonpolar, bipolar, multisite were calculated the proportion were counted. ± indicates the standard deviations from three biology repeats. NA, not available.

**Table 1 microorganisms-09-00495-t001:** The bacterial strains and plasmids used in this study.

Strain or Plasmid	Relevant Characteristics ^a^	Source or Reference
Strains		
*Escherichia coli*		
DH5α	*F^-^φ80(lacZ)ΔlacX74hsdR(r_k_^−^, m_k_^+^)ΔrecA1398endA1tonA*	TransGene Biotech, Beijing, China
BL21	F^-^ omp T hsdS (r_B_^−^ m_B_^−^)gal dcm (DE3)	TransGene Biotech, Beijing, China
*Xanthomonas. oryzae* pv. *oryzae*		
PXO99^A^	Wild-type strain, Philippine race 6, Cp^r^	[32]
∆*gdpX6*	g*dpX6* gene deletion mutant derived from PXO99^A^, Cp^r^	This study
Plasmid		
pKMS1	Suicidal vector carrying *sacB* gene for mutagenesis, Km^r^	[33]
pK*gdpX6*	pKMS1 with *gdpX6*, Km	This study
pColdSUMO	Protein expression vector with N-terminal SUMO-His_6_-tag, Ap^r^	Haigene Biotech, Harbin, China
pCGdpX6^GGDEF^	pColdSUMO carrying the coding sequence of the GGDEF domain (318 to 495 aa) of GdpX6, Ap^r^	This study
pCGdpX6^GGDEF-E411A^	pColdSUMO carrying the coding sequence for the point mutation of E^411^ in GGDEF domain of *gdpX6,* Ap^r^	This study
pCGdpX6^GGDEF-D403A^	pColdSUMO carrying the coding sequence for the point mutation of D^403^ in GGDEF domain of *gdpX6,* Ap^r^	This study
pETPleD	pET28b containing the PleD coding sequence, Km^r^	[34]
pET28b	pET28b protein expression vector, Km^r^	Laboratory collection
pB*gfp*	Broad-host range expression vector pBBR1MCS-4 carrying *gfp*, Ap^r^	[35]
pB*gdpX6gfp*	pB*gfp* carrying full-length *gdpX6*, Ap^r^	This study
pB*gdpX6^E411A^gfp*	pB*gfp* carrying full-length *gdpX6* with E^411^ point mutation, Ap^r^	This study
pB*gdpX6^D403A^gfp*	pB*gfp* carrying full-length *gdpX6* with D^403^ point mutation, Ap^r^	This study

^a^ Km^r^, Ap^r^ and Cp^r^ indicate resistance to kanamycin, ampicillin, and gentamicin, respectively.

## Data Availability

Not applicable.

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
