# Peer review of "Diguanylate Cyclase GdpX6 with c-di-GMP Binding Activity Involved in the Regulation of Virulence Expression in Xanthomonas oryzae pv. oryzae"

_microorganisms, 2021, doi:10.3390/microorganisms9030495_

Round 1

Reviewer 1 Report

In general, the manuscript is well structured and well written. However I would like to make a few recommendations.

General comments on grammar/spelling:

  • pay attention to the grammar (e.g., line 43, like instead of liking; line 44, contribute instead of are contribute; line 67, contain instead of containing. etc.)
  • avoid the use of the if introducing a concept for the first time (e.g., line 53, the biofilm formation)
  • avoid the use of acronyms if they are not further used in the text (e.g., lines 160 and 175).
  • explain the acronym when using it for the first time (lines 36, 402 and 403)
  • Line 478, I think you meant however instead of moreover.

Comments on the methods/results/discussion:

  • Section 2.4, mention that you use PleD as control.
  • Sections 2.8 and 2.11, how many replicates did you use for virulence and EPS assays? Include this information also in the figure legends.
  • I recommend including a section summarizing the statistical methods.
  • Is increased virulence in ΔgdpX6 related to increased motility?
  • Relationship between T4P and and DGC should be better discussed, the following paper of T4P in Lysobacter might be useful 10.1128/AEM.03397-16
  • Lines 67-77 are difficult to follow.
  • Table 1, I recommend aligning the text left to make it simpler and more friendly.
  • Line 104, might be worthy to give the website of NCBI.
  • Line 237, might bind c-di-GMP. It was demonstrated later on, avoid the use of can.
  • Line 251, mention the organism where you expressed the mutagenized proteins.
  • Line 265, mention if the activity of SUMOHis6-GdpX6… was stronger or lower than the wild type.
  • Figure 2a, are the first three “treatments” controls? Please, include this information in the legend. Bc3-5/pETPleD not significantly different?
  • Figure 2b, what is BSA?
  • Lines 306-307, over expression of GdpX6… significantly decreased virulence.
  • Figure 4, t-test using PXO99A as reference?
  • Line 325, the term restored is not recommended here as activity was not decreased/abolished in the mutant deficient of gdpX6.
  • Line 441, which are the previous studies? Include references. If referring to the results of your study you should say, based on the above results.
  • Line 422, I would include some notes on site-A in the discussion.
  • Line 430, a citation is missing.
  • Lines 436-439, difficult to follow.
  • Line 482, what is the implication of Cache?

Author Response

We would like to thank you for your constructive comments. In this letter, we have addressed the reviewer’s concerns, and pointed out where and what improvements or revisions, where applicable, have been made to the manuscript. And we checked and modified the phrasing and grammar issues through the manuscript.

General comments on grammar/spelling:

  1. pay attention to the grammar (e.g., line 43, like instead of liking; line 44, contribute instead of are contribute; line 67, contain instead of containing. etc.)

Response: Thank you for reviewer’s suggestion. We modified the words following reviewer’s suggestion. Please see Line 43, Line 44 and Line 66.

  1. avoid the use of the if introducing a concept for the first time (e.g., line 53, the biofilm formation) avoid the use of acronyms if they are not further used in the text (e.g., lines 160 and 175).

Response: We modified wrong uses of “the” throughout manuscript and the use of acronyms of some words which did not further mentioned in text. Please see Line 53, Line 164 and Line180.

  1. explain the acronym when using it for the first time (lines 36, 402 and 403)

Response: We added the explanation for acronym. Please see Line 35. The explanation for acronym of Xcc, Xac and Xoc are described in Line 249-250.

  1. Line 478, I think you meant however instead of moreover.

Response: We modified “Moreover” to “However” following reviewer’s suggestion. Please see Line 515.

Comments on the methods/results/discussion:

  1. Section 2.4, mention that you use PleD as control.

Response: We mentioned as “The E. coli strains containing pETPleD/BC3-5 RNA and pET28b/BC3-5 RNA were used as positive control and its corresponding negative control, respectively” in Materials and Methods. Please see Line 131-133.

  1. Sections 2.8 and 2.11, how many replicates did you use for virulence and EPS assays? Include this information also in the figure legends.

Response: We used ten leaves and three replicates in the assays of virulence and EPS, respectively. We add the description about replicates in Figure legend and Materials and Methods. Please see Line 192, Line 219-220, Line 344-345, Line 539.

  1. I recommend including a section summarizing the statistical methods.

Response: We thank reviewer’s comment. We added a paragraph as “Statistical methods” in Material and methods. Please see Line 237-241.

  1. Is increased virulence in ΔgdpX6 related to increased motility?

Response: At present, there is no evidence to prove that there is a direct relationship between swimming or sliding motility and virulence in Xoo. We supposed that deletion of the active DGC gdpX6 might affect the intracellular level of c-di-GMP to regulate some pathways like motility, biofilm formation, and the expression of downstream virulence factors. We discussed how gdpX6 regulate the pilus-depending motility in Xoo.

  1. Relationship between T4P and and DGC should be better discussed, the following paper of T4P in Lysobacter might be useful 10.1128/AEM.03397-16

Response: Thanks for the reviewer’s good suggestion. We further discussed the relationship between T4P and DGC in Discussion. Please see Line 488-491.

  1. Lines 67-77 are difficult to follow.

Response: A sentence was modified in the revised MS: “Additionally, the RXXD motif of the allosteric inhibition site (I site) which is located before the A site in GGDEF domain participates in c-di-GMP binding”. Please see Line 64-66.

  1. Table 1, I recommend aligning the text left to make it simpler and more friendly.

Response: We modified the Table 1 following the reviewer’s good suggestion. Please see Table 1.

  1. Line 104, might be worthy to give the website of NCBI.

Response: We added the website of NCBI following the reviewer’s good suggestion. Please see Line 104.

  1. Line 237, might bind c-di-GMP. It was demonstrated later on, avoid the use of can.

Response: The sentence was changed to “GdpX6 might function as an active DGC and bind with c-di-GMP”. Please see Line 256-257.

  1. Line 251, mention the organism where you expressed the mutagenized proteins.

Response: We added “E. coli BL21(DE3)” which we expressed the mutagenized proteins. Please see Line 277.

  1. Line 265, mention if the activity of SUMOHis6-GdpX6… was stronger or lower than the wild type.

Response: We changed the sentence as “However, the value of RFI decreased about 81% as compared to GdpX6GGDEF (Figure 2a), suggestive of the lower DGC activity of GdpX6GGDEF-E411A as compared to GdpX6GGDEF.” in the revised MS. Please see Line 284-286.

  1. Figure 2a, are the first three “treatments” controls? Please, include this information in the legend. Bc3-5/pETPleD not significantly different?

Response: In Figure 2a, The E. coli strains containing pETPleD/BC3-5 RNA and pET28b/BC3-5 RNA were used as positive control and its corresponding negative control, respectively. The E. coli strain containing the empty vector pColdSUMO and BC3-5 RNA was used as the negative control of pColdGdpX6GGDEF/BC3-5 RNA. We add related description in manuscript. Please see Line 131-133. We added the significance analysis of pETPleD/BC3-5 RNA and pET28b/BC3-5, pColdSUMO /BC3-5 RNA and pColdGdpX6GGDEF /BC3-5. Please see Figure 2a.

  1. Figure 2b, what is BSA?

Response: We used SUMO-His6 and Bovine Serum Albumin (BSA) as negative controls. We deleted BSA for making the reader easier following the manuscript. Please see Figure 2b.

  1. Lines 306-307, over expression of GdpX6… significantly decreased virulence.

Response: The sentence was changed to “overexpression of GdpX6 or GdpX6D403A in PXO99A significantly decreased the virulence (by about 25%) in comparison to PXO99A (P < 0.05)”. Please see Line 335-336.

  1. Figure 4, t-test using PXO99A as reference?

Response: Yes, we compared the virulence of the mutant ∆gdpX6 strain, the complement strains of ∆gdpX6 and gdpX6-overexpressing strains with that of PXO99A, respectively.

  1. Line 325, the term restored is not recommended here as activity was not decreased/abolished in the mutant deficient of gdpX6.

Response: We changed the word “restored” to “recovered”. Please see Line 354.

  1. Line 441, which are the previous studies? Include references. If referring to the results of your study you should say, based on the above results.

Response: We changed the sentence to “Based on the studies carried on the homologs of GdpX6 (1)”. Please see Line 443.

  1. Line 422, I would include some notes on site-A in the discussion.

Response: We discussed the A site in GGDEF domain in discussion. Please see Line 448-449.

  1. Line 430, a citation is missing.

Response: We thank reviewer’s comment. We added the related reference. Please see Line 458.

  1. Lines 436-439, difficult to follow.

Response: We rewrote the sentence: Moreover, flagellum-dependent biofilm regulatory response can be induced through the elimination of flagellum, which can improve the level of c-di-GMP and enhances the biofilm formation, and this response requires at least three specific DGCs in the V. cholerae. Please see Line 468-471.

  1. Line 482, what is the implication of Cache?

Response: We changed the word Cache to Cache_1 domain which was described in Results. Please see Line 245 and Line 517.

Reviewer 2 Report

The manuscript “Diguanylate cyclase GdpX6 with c-di-GMP binding activity involved in the regulation of virulence expression in Xanthomonas oryzae pv. oryzae” by Yan et al. characterizes the diguanylate cyclase GdpX6. First, the authors demonstrate that GdpX6 is indeed an active diguanylate cyclase using in vitro and in vivo analyses. Second, the authors show that GdpX6 is important for regulation of virulence, biofilm formation and motility and that the diguanylate cyclase activity of GdpX6 is critical for its biological function.

The authors used GFP-fused GdpX6 for complementation and overexpression analyses, which I find a bit problematic. Lack of complementation by catalytically inactive version of GdpX6 support that the GFP fusion did not cause any dominant negative effect. However, I suggest the authors should explain why they chose to use GFP fusions for these purposes (I do not request any experiments, just a statement in the discussion portion). Other that this issue the experiments appear to be solid.

The manuscript contains a significant number of phrasing and grammar issues that make it very unfriendly to the readers. I stated specific issues I observed in my comments below. Nevertheless, I strongly suggest that the authors should review their text at least one more time.

I addition the number of biological repeats that were conducted in some of the experiments are not mentioned in the text.

*General issues and clarifications

  1. The authors should provide a good explanation why they used GFP fused protein for their complementation and over expression experiments. GFP is a large tag that harbor folding issues when transported into the periplasm. It is not an ideal tag when it comes for overexpression of sec-dependent lipoproteins. This is problematic since fusing the protein to large tag as GFP might affect protein structure, activity, and the interaction with partner proteins.
  2. Sections 2.8. and 2.11. : The authors should state how many biological repeats were used in each experiment.
  3. Section 2.9 and 3.5: were different media used for the swimming and sliding experiments? If so, then the authors cannot conclude that the mutant is affecting one pathway and not the other since the metabolic state of the cells is different.
  4. section 3.1: The authors should state the locus tag number of GdpX6. If there is no deposit available, the authors should deposit the sequence themselves into NCBI and refer the readers to the deposit number from the text.
  5. section 3.2 (lines 248-260): Please make it clear to the readers that described experiments were conducted in E. coli.
  6. Line 261: please make it clear to the readers that the experiments were conducted using a purified protein.
  7. Line 379: Please add the western blot analysis as a supplementary figure.
  8. Lines 402-405: the authors state that the GdpX6 homolog XCC2731 control extracellular hydrolases. Have authors checked hydrolase activity in the gdpX6 mutant? If not, why test all the other reported phenotypes but not this one?
  9. I suggest the author should avoid the phrase “flagellar motility” since they never tested whether the flagellum is involved in the observed phenotypes. It is better to just use the term “motility” instead.
  10. As far as I know the observance of crystal violet diluted in ethanol is read at 570 nm and not 490 nm. Why was this wavelength used?

*grammar and wording issues:

Line 14: Cyclic diguanylate monophosphate (c-di-GMP) is a second messenger present in bacteria, not only the phytopathogenic ones

Lines 16 and 17: “In the genome of Xanthomonas oryzae pv. oryzae, the causal agent of bacterial blight of rice, 11 genes encode single GGDEF domain proteins” can be phrased better.

Lines 43 and 424– please change “liking” to “like”

Line 44 – please delete the word “are”

Line 48, 50 and 54– please delete the word “the”

Line 57 - please delete the word “or”

Line 67 – Please change “the” to “a”

Line 68 and 69 – Please change “from” to “of”

Line 72,76 and 90 – please delete the word “the”

Line 81 and 87: please change “identified” to “characterize”

Line 90: Please change “indicated that GdpX6 protein localizes” to “identified that GdpX6 protein is localizes”

Line 92: Please change “the virulence phenotypes in Xoo.” to “virulence in Xoo”

Line 121-122. The sentence is not clear. Please rephrase.

Line 133: Please delete the word “these”

Line 230: Please change “X. axonopodis pv. citri” to “X. citri pv. citri”

Line 240: The sentence is not clear. Please rephrase.

Line 259: Please change “with” to “as the”

Lines 296 and 297: The sentence is not grammatically correct. Please rephrase.

Lines 299-300: Please change “and longer lesion lengths increased about 20% as compared to the wildtype strain PXO99A” to “and legion length was increased in about 20% compared to the wildtype strain”

Lines 302-304: it is unnecessary to mention that D403A and E411A are point mutations. These mutants were mentioned earlier in the text.

Lines 433: Please delete the word “the”

Line 437: Please rephrase “eliminate flagellum”

Line 441:  I am not sure what “consistent with the active DGCs” is referring to. Do the authors refer to the catalytic activity of the protein or that DGCs in other organisms display the same phenotype? Please be clear.

Line 442: Please change “motive” to “motile”

Line 449: please change “The c-di-GMP signaling especially diverse c-di-GMP receptors” to “c-di-GMP signaling and specific c-di-GMP receptors” or to “c-di-GMP signaling”

Line 456: Please change “it was GdpX6 was found to acts” to “GdpX6 was found to act”

Line 457: please change “it” to “this”

Line 225: Please change “protein domain Cache”  to “Cache protein domain”

Line 234: Please change “amino sequence” to “amino acid sequence”

Line 250: Please change “of GGDE411F” to “of the GGDE411F”

Line 261: Please change “enzyme activities of GdpX6 were” to “enzyme activity of GdpX6 was”

Line 276: Please change “The bioinformatics” to “bioinformatic”

Line 278: Please state which proteins “the proteins” refer to, it its only GdpX6 that it should be “the protein”

Line 295: Please change “the” to “a”

Line 307: what was “was significantly decreased”? The authors should specifically state that they are referring to lesion length

Lines 328 and 329: I am assuming the statement “When the E411 or D403 were mutated in GdpX6, the influence of GdpX6 on the swimming motility disappeared” is referring to the overexpression strain. Please be clear about it

Line 369: what is “colony surface checking”?

Section 2.3 is describing standard cloning and mutagenesis procedures. This paragraph is very hard to follow. The authors should consider just citing a site-directed mutagenesis protocol and refer the readers to table S1 for the primers.

*Technical issues with figures and tables:

All figures: resolution is not good. Please improve picture quality.

Table 1: PXO99A – please change the reference from “lab collection” to the genome sequencing publication of this strain.

Table 1: gdpX6 – please add the locus tag in brackets.

Figure 1 legend: Please state all the locus tags of all the proteins which were used in the sequence alignment analysis.

Figure 5 legend: was the presented data represents nine repeats or three repeats? The wording is confusing.

Author Response

Dear reviewer,

We would like to thank you for your constructive comments. In this letter, we have addressed the reviewer’s concerns, and pointed out where and what improvements or revisions, where applicable, have been made to the manuscript. And we checked and modified the phrasing and grammar issues through the manuscript.

Comments and Suggestions for Authors

  1. The authors used GFP-fused GdpX6 for complementation and overexpression analyses, which I find a bit problematic. Lack of complementation by catalytically inactive version of GdpX6 support that the GFP fusion did not cause any dominant negative effect. However, I suggest the authors should explain why they chose to use GFP fusions for these purposes (I do not request any experiments, just a statement in the discussion portion). Other that this issue the experiments appear to be solid.

The authors should provide a good explanation why they used GFP fused protein for their complementation and over expression experiments. GFP is a large tag that harbor folding issues when transported into the periplasm. It is not an ideal tag when it comes for overexpression of sec-dependent lipoproteins. This is problematic since fusing the protein to large tag as GFP might affect protein structure, activity, and the interaction with partner proteins.

Response: Thanks for reviewer’s good comments. Firstly, we found that in trans expression of the full length gdpX6 without the GFP tag in ∆gdpX6 restored the virulence phenotypes. For analysis of the subcellular localization of GdpX6, C-terminal fusions of green fluorescent protein (GFP) to GdpX6 was generated. Western blotting analysis showed that GdpX6-GFP fusion protein expressed in ∆gdpX6(pBgdpX6gfp) and ∆gdpX6(pBgdpX6gfp) exhibited the similar virulence phenotypes as that of wildtype PXO99A. Those results suggest GdpX6-GFP displayed similar biological function as GdpX6, therefore we used GFP fused protein for complementation and over-expression experiments in this study. We added some descriptions in manuscript. Please see Line 323-326 and Lin 334-335.

  1. The manuscript contains a significant number of phrasing and grammar issues that make it very unfriendly to the readers. I stated specific issues I observed in my comments below. Nevertheless, I strongly suggest that the authors should review their text at least one more time.

Response: Thanks for the reviewer’s good suggestion. We modified the specific issues following the reviewer’s suggestions. And we checked and modified the phrasing and grammar issues through the manuscript.

  1. I addition the number of biological repeats that were conducted in some of the experiments are not mentioned in the text.

Response: We add the description about replicates in Figure legend and Materials and Methods. Please see Line 192, Line 219-220, Line 344-345, Line 538.

*General issues and clarifications

  1. Sections 2.8. and 2.11.: The authors should state how many biological repeats were used in each experiment.

Response: We add the description about replicates in Figure legend and Materials and Methods. Please see Line 192, Line 219-220, Line 344-345, Line 538.

  1. Section 2.9 and 3.5: were different media used for the swimming and sliding experiments? If so, then the authors cannot conclude that the mutant is affecting one pathway and not the other since the metabolic state of the cells is different.

Response: As we described in 2.9 and 3.5, we used semi-solid plates containing 0. 25% agar for testing swimming assay and SB medium plates containing 0.6% agar for analyzing sliding motility. Please see Line 198-199, Line 349-350.

  1. section 3.1: The authors should state the locus tag number of GdpX6. If there is no deposit available, the authors should deposit the sequence themselves into NCBI and refer the readers to the deposit number from the text.

Response: We stated the locus tag number of GdpX6 as PXO_02019 (here GdpX6 [GGDEF-domain protein of Xoo 6] in introduction, therefore we directly used GdpX6 in the following text. The GenBank accession number of GdpX6 protein was added to the Figure 1 legend. Please see Line 84 and Line 260.

  1. section 3.2 (lines 248-260): Please make it clear to the readers that described experiments were conducted in coli.

Response: Thanks for the reviewer’s good suggestion. We described experiments were conducted in “E. coli BL21(DE3)”. Please see Line 277.

  1. Line 261: please make it clear to the readers that the experiments were conducted using a purified protein.

Response: We added the purified proteins as “the enzyme activity of purified SUMOHis6-GdpX6GGDEF, SUMOHis6-GdpX6GGDEF-E411A and SUMOHis6 proteins was tested using GTP as substrate and the concentration of c-di-GMP in the reaction was detected by LC-MS/MS.”. Please see Line 287-288.

  1. Line 379: Please add the western blot analysis as a supplementary figure.

Response: We added the results of western blot analysis as Figure S2. Please see Figure S2.

  1. Lines 402-405: the authors state that the GdpX6 homolog XCC2731 control extracellular hydrolases. Have authors checked hydrolase activity in the gdpX6 mutant? If not, why test all the other reported phenotypes but not this one?

Response: Thanks for the reviewer’s good suggestion. We detected whether GdpX6 affect the extracellular hydrolases of Xoo. The results showed that GdpX6 did not regulate the extracellular enzymatic activities of Xoo. We added the “2.12 Extracellular enzymatic activity assay” in Material and Methods, “3.7. GdpX6 does not control EPS production and extracellular enzymatic activities of Xoo” in Results. We discussed the results in discussion. Please see Line 221-231, Line 403-407, Line 438, Line 540-544 and Figure S4.

  1. I suggest the author should avoid the phrase “flagellar motility” since they never tested whether the flagellum is involved in the observed phenotypes. It is better to just use the term “motility” instead.

Response: We used “swimming motility” instead of “flagellar motility” in manuscript.

  1. As far as I know the observance of crystal violet diluted in ethanol is read at 570 nm and not 490 nm. Why was this wavelength used?

Response: The biofilm formation assay has been described in many articles (An et al, 2010; Takahashi et al, 2000; Xue et al, 2018). As described in the article, the biofilm was dissolved in ethanol, and the absorbance was detected at 490 nm. So we tested biofilm formation by using this method.

Reference:

An, S.; Wu, J.; Zhang, L. H. Modulation of Pseudomonas aeruginosa biofilm dispersal by a cyclic-di-GMP phosphodiesterase with a putative hypoxia-sensing domain. Appl. Environ. Microbiol. 2010, 76, 8160-8173.

Takahashi K, Nei M. Efficiencies of fast algorithms of phylogenetic inference under the criteria of maximum parsimony, minimum evolution, and maximum likelihood when a large number of sequences are used. Mol Biol Evol. 2000,17:1251–1258.

Xue, D. R.; Tian, F.; Yang, F. H.; Chen, H. M.; Yuan, X.; Yang, C. H.; He, C. Y. Phosphodiesterase EdpX1 promotes Xanthomonas oryzae pv. oryzae virulence, exopolysaccharide production, and biofilm formation. Appl. Environ. Microbiol. 2018, 84, e01717-18.

*grammar and wording issues:

Line 14: Cyclic diguanylate monophosphate (c-di-GMP) is a second messenger present in bacteria, not only the phytopathogenic ones

Response: We changed the sentence as “Cyclic diguanylate monophosphate (c-di-GMP) is a second messenger present in bacteria.”. Please see Line 14.

Lines 16 and 17: “In the genome of Xanthomonas oryzae pv. oryzae, the causal agent of bacterial blight of rice, 11 genes encode single GGDEF domain proteins” can be phrased better.

Response: We rewrote the sentence as “In the genome of Xanthomonas oryzae pv. oryzae, the causal agent of bacterial blight of rice, there are 11 genes which encode single GGDEF domain proteins”. Please see Line 16-18.

Lines 43 and 424– please change “liking” to “like”

Response: We changed “liking” to “like”. Please see Line 43 and Line 457.

Line 44 – please delete the word “are”

Response: We deleted the word “are”. Please see Line 44.

Line 48, 50 and 54– please delete the word “the”

Response: We deleted the word “the”. Please see Line 48, Line 50 and Line 53.

Line 57 - please delete the word “or”

Response: We changed the word “or” to “and”. Please see Line 57.

Line 67 – Please change “the” to “a”

Response: We changed “the” to “a”. Please see Line 66.

Line 68 and 69 – Please change “from” to “of”

Response: We changed “from” to “of”. Please see Line 67 and Line 68.

Line 72,76 and 90 – please delete the word “the”

Response: We deleted the word “the”. Please see Line 71, 75 and Line 89.

Line 81 and 87: please change “identified” to “characterize”

Response: We changed “identified” to “characterize”. Please see Line 80 and Line 86.

Line 90: Please change “indicated that GdpX6 protein localizes” to “identified that GdpX6 protein is localizes”

Response: We changed “indicated that GdpX6 protein localizes” to “identified that GdpX6 protein has multisite distribution in the cell”. Please see Line 89-90.

Line 92: Please change “the virulence phenotypes in Xoo.” to “virulence in Xoo

Response: We changed “the virulence phenotypes in Xoo” to “the virulence in Xoo”. Please see Line 91.

Line 121-122. The sentence is not clear. Please rephrase.

Response: We rewrote the sentence as “Expression of target proteins was induced by addition of isopropyl β- d-1-thiogalactopyranoside (IPTG)at the final concentration of 0.5 mM. Then, the bacterial cultures were incubated for 16 h at 16°C.”. Please see Line 122-124.

Line 133: Please delete the word “these”

Response: We deleted the word “these”. Please see Line 136.

Line 230: Please change “X. axonopodis pv. citri” to “X. citri pv. citri

Response: We changed “X. axonopodis pv. citri” to “X. citri pv. citri”. Please see Line 250.

Line 240: The sentence is not clear. Please rephrase.

Response: We rewrote the sentence as “The numbers represent the start and end amino acids of the predicted domains according to NCBI’s conserved domain database and the SMART database”. Please see Line 260-261.

Line 259: Please change “with” to “as the”

Response: We changed “with” to “as the”. Please see Line 284.

Lines 296 and 297: The sentence is not grammatically correct. Please rephrase.

Response: We rewrote the sentence as “The bacterial cells were inoculated onto the leaves of susceptible rice plants by leaf-clipping method”. Please see Line 325-326.

Lines 299-300: Please change “and longer lesion lengths increased about 20% as compared to the wildtype strain PXO99A” to “and legion length was increased in about 20% compared to the wildtype strain”

Response: We changed the sentence to “the lesion lengths were increased by about 20% as compared to wildtype strain PXO99A”. Please see Line 328-329.

Lines 302-304: it is unnecessary to mention that D403A and E411A are point mutations. These mutants were mentioned earlier in the text.

Response: We deleted related sentences. Please see Line 331.

Lines 433: Please delete the word “the”

Response: We deleted the word “the”. Please see Line 467.

Line 437: Please rephrase “eliminate flagellum”

Response: We rewrote the sentence as “flagellum-dependent biofilm regulatory response can be induced through the elimination of flagellum, which can improve the level of c-di-GMP and enhances the biofilm formation, and this response requires at least three specific DGCs in the V. cholerae”. Please see Line 468-471.

Line 441:  I am not sure what “consistent with the active DGCs” is referring to. Do the authors refer to the catalytic activity of the protein or that DGCs in other organisms display the same phenotype? Please be clear.

Response: We added description and references about the active DGCs as “like PleD or WspR (2-4)”. Please see Line 457-458.

Line 442: Please change “motive” to “motile”

Response: We changed “motive” to “motile”. Please see Line 474.

Line 449: please change “The c-di-GMP signaling especially diverse c-di-GMP receptors” to “c-di-GMP signaling and specific c-di-GMP receptors” or to “c-di-GMP signaling”

Response: We changed “The c-di-GMP signaling especially diverse c-di-GMP receptors” to “The c-di-GMP signaling and specific diverse c-di-GMP receptors”. Please see Line 481-482.

Line 456: Please change “it was GdpX6 was found to acts” to “GdpX6 was found to act”

Response: We changed “it was GdpX6 was found to acts” to “GdpX6 was found to act”. Please see Line 491.

Line 457: please change “it” to “this”

Response: We changed “it” to “this”. Please see Line 492.

Line 225: Please change “protein domain Cache” to “Cache protein domain”

Response: We changed “protein domain Cache” to “Cache_1 protein domain”. Please see Line 345.

Line 234: Please change “amino sequence” to “amino acid sequence”

Response: We changed “amino sequence” to “amino acid sequence”. Please see Line 258.

Line 250: Please change “of GGDE411F” to “of the GGDE411F”

Response: We changed “of GGDE411F” to “of the GGDE411F”. Please see Line 274.

Line 261: Please change “enzyme activities of GdpX6 were” to “enzyme activity of GdpX6 was”

Response: We changed “enzyme activities of GdpX6 were” to “the enzyme activity of purified SUMOHis6-GdpX6GGDEF, SUMOHis6-GdpX6GGDEF-E411A and SUMOHis6 proteins was”. Please see Line 286-287.

Line 276: Please change “The bioinformatics” to “bioinformatic”

Response: We changed the word “The bioinformatics” to “bioinformatic”. Please see Line 303.

Line 278: Please state which proteins “the proteins” refer to, it its only GdpX6 that it should be “the protein”

Response: We added the proteins which were used in ITC assay. Please see Line 305.

Line 295: Please change “the” to “a”

Response: We changed “the” to “a” following reviewer’s suggestion. Please see Line 322.

Line 307: what was “was significantly decreased”? The authors should specifically state that they are referring to lesion length

Response: We rewrote the sentence as “the lesion lengths were increased by about 20% as compared to wildtype strain PXO99A”. Please see Line 336.

Lines 328 and 329: I am assuming the statement “When the E411 or D403 were mutated in GdpX6, the influence of GdpX6 on the swimming motility disappeared” is referring to the overexpression strain. Please be clear about it

Response: We rewrote the sentence as “When the E411 or D403 were mutated in GdpX6, the influence of overexpression of gdpX6 in PXO99A on the swimming motility disappeared”. Please see Line 358-359.

Line 369: what is “colony surface checking”?

Response: We changed the words “colony surface checking” to “colony examining”. Please see Line 399.

Section 2.3 is describing standard cloning and mutagenesis procedures. This paragraph is very hard to follow. The authors should consider just citing a site-directed mutagenesis protocol and refer the readers to table S1 for the primers.

Response: We rewrote the sentence about cloning and mutagenesis procedures in this paragraph. Please see Line 110-115.

*Technical issues with figures and tables:

All figures: resolution is not good. Please improve picture quality.

Response: We added the figure with high resolution in manuscript.

Table 1: PXO99A – please change the reference from “lab collection” to the genome sequencing publication of this strain.

Response: We added the related reference of PXO99A instead of “lab collection”. Please Table 1.

Table 1: gdpX6 – please add the locus tag in brackets.

Response: We added the statement of the locus tags of GdpX6. Please Table 1.

Figure 1 legend: Please state all the locus tags of all the proteins which were used in the sequence alignment analysis.

Response: Thanks for the reviewer’s good suggestion. We added the statement of the locus tags of all the proteins which were used in the sequence alignment analysis. Please see Line 267-272.

Figure 5 legend: was the presented data represents nine repeats or three repeats? The wording is confusing.

Response: We added the description about the data repeats as “The error bars represented standard deviations of the lesion lengths from ten leaves. Three independent experiments were performed with similar results”. Please see Line 344-345.